# Learning to Quantize for Training Vector-Quantized Networks

**Peijia Qin** [1]   **Jianguo Zhang** [1] [2]

## Abstract

Deep neural networks incorporating discrete latent variables have shown significant potential in sequence modeling. A notable approach is to leverage vector quantization (VQ) to generate discrete representations within a codebook. However, its discrete nature prevents the use of standard backpropagation, which has led to challenges in efficient codebook training. In this work, we introduce **Meta-Quantization (MQ)**, a novel vector quantization training framework inspired by meta-learning. Our method separates the optimization of the codebook and the auto-encoder into two levels. Furthermore, we introduce a hyper-net to replace the embedding-parameterized codebook, enabling the codebook to be dynamically generated based on the feedback from the auto-encoder. Different from previous VQ objectives, our innovation results in a meta-objective that makes the codebook training task-aware. We validate the effectiveness of MQ with VQVAE and VQ-GAN architecture on image reconstruction and generation tasks. Experimental results showcase the superior generative performance of MQ, underscoring its potential as a robust alternative to existing VQ methods.

## 1. Introduction

Learning discrete latent variables is favorable for tasks naturally modeled as sequences of discrete symbols, including language and speech (Vinyals et al., 2015). Vector-quantized networks (VQNs) provide an effective approach for this by

[1]Research Institute of Trustworthy Autonomous Systems and Department of Computer Science and Engineering, Southern University of Science and Technology, Shenzhen 518055, China. [2]Guangdong Provincial Key Laboratory of Brain-inspired Intelligent Computation, Department of Computer Science and Engineering, Southern University of Science and Technology, Shenzhen 518055, China, and also with the Pengcheng Laboratory. Correspondence to: Jianguo Zhang <zhangjg@sustech.edu.cn>.

*Proceedings of the 42nd International Conference on Machine Learning*, Vancouver, Canada. PMLR 267, 2025. Copyright 2025 by the author(s).

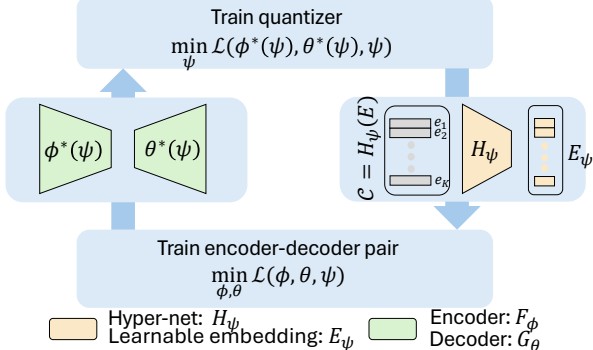

*Figure 1.* **Our framework is based on bi-level optimization.** Meta-quantization learns the codebook and encoder-decoder pair using a bi-level optimization framework. At the lower level, the encoder-decoder pair is trained to converge while keeping the codebook fixed. At the upper level, the codebook is optimized via hyper-gradient descent using the optimal encoder-decoder pair.

learning latent variables through vector quantization (VQ, Gray (1984)), a process that quantizes features into clusters referred to as codes. The concept of VQNs was first introduced with the vector-quantized variational autoencoder (VQVAE, van den Oord et al. (2017)) in the context of generative models. Subsequent research has demonstrated that training autoregressive priors on discrete representations obtained via vector quantization yields highly effective models for image generation (Razavi et al., 2019; Roy et al., 2018; Ramesh et al., 2021; Esser et al., 2021; Chang et al., 2023). Beyond image generation, VQNs have achieved notable success in speech generation (Dhariwal et al., 2020) and have extended their utility to other domains, such as image representation learning (Caron et al., 2020) and speech representation learning (Chung et al., 2020).

The training approach for vector-quantized networks (VQNs), as implemented in VQVAE, involves learning a codebook $\mathcal{C}$ to represent compressed semantic data. The encoder $F_\phi$ maps input data to an embedding, which is subsequently quantized by selecting the nearest neighbor in $\mathcal{C}$. The selected code replaces the embedding and is then passed to the decoder $G_\theta$ to produce the output. Since the quantization operation introduces a non-differentiable bottleneck, a straight-through estimator (STE, Bengio et al. (2013)) is employed to enable gradient flow through the

VQ layer to the encoder during backpropagation. However, because the backpropagated gradient bypasses the codebook in the STE framework, the codebook is instead optimized using a vector quantization objective. This objective aligns the distribution of the encoder embeddings with the selected codes to achieve distribution matching.

Despite its effectiveness, the current training framework for vector quantization faces several challenges. First, vector quantization often suffers from a phenomenon known as index collapse, where only a small subset of codes remains active during training (Kaiser et al., 2018). This issue arises because frequently optimized token embeddings gradually align more closely with the feature map distributions, while less frequently or never-optimized token embeddings (inactive tokens) are excluded from the training process. These inactive tokens remain unused during inference, leading to suboptimal codebook utilization. Second, the codebook is optimized solely using the vector quantization objective, which focuses on distribution matching. However, due to the straight-through estimator (STE), gradients from the task loss bypass the codebook during backpropagation. As a result, codebook updates are agnostic to the specific task, potentially undermining its effectiveness for the overall objective of image modeling.

Drawing inspiration from meta-learning (Finn et al., 2017), we propose a novel vector quantization training framework termed **M**eta **Q**uantization (MQ). Our method builds directly on the vector quantization mechanism and reparameterizes the codebook by introducing a separate *hyper-net* $H_\psi$. The hyper-net predicts the parameter matrices of the codebook from a learned embedding. While maintaining the same expressivity at the quantization bottleneck, the hyper-net aggregates gradient feedback from all matched codes and generates the codebook holistically. This design accelerates convergence without introducing additional complexity to the quantization operation and enables seamless integration with existing approaches. The theoretical advantages (for example, accelerating convergence) of such an overparameterized architecture have been studied in Arora et al. (2018). Moreover, we formulate an asymmetric bi-level optimization problem to allow the hyper-net to learn quantization *in a meta-learning fashion*. In this framework, the hyper-net functions as a meta-learner (analogous to the hyperparameters in meta-learning), while the encoder-decoder pair acts as a task-specific learner. As illustrated in Figure 1, the hyper-net and the encoder-decoder pair are optimized at the upper and lower levels, respectively. At the upper level, the hyper-net anticipates the encoder-decoder pair's future performance by tentatively training them until convergence (unrolling for one step as a practical surrogate) while keeping the codebook fixed. Subsequently, the hyper-net is updated via hyper-gradient descent to minimize the loss for one step, leveraging the optimal encoder and decoder as

functions of the hyper-net. At the lower level, the tentative optimization steps of the encoder-decoder pair from the first stage are undone, as a better hyper-net is identified in the previous step. Instead, the encoder and decoder are optimized for one step using gradient descent with the updated hyper-net. These two levels of optimization are performed iteratively until convergence.

Figure 2 provides a detailed illustration of the training procedure and gradient flow in our proposed framework. Specifically, the meta-objective is implemented by unrolling the lower-level training for several steps, with its gradient computed through hypergradient descent (i.e., optimizing through the inner gradient descent path). The hypergradient paths demonstrate how the gradient from the task loss can now reach the codebook through multiple routes. This contrasts with previous codebook training strategies, where the task loss has no direct gradient impact on the codebook. A practical advantage of our framework is that the hyper-net can be discarded after training, retaining only the generated weights of the codebook. This means the downstream does not suffer from additional inference burden. Furthermore, the bi-level training procedure does not affect downstream tasks, as the parameters trained in the first stage are fixed. This makes meta-learned codebooks as efficient as their conventional counterparts.

We evaluate the effectiveness of our framework across the VQVAE and VQGAN architectures on standard image reconstruction and generation tasks. The results show that the MQ approach consistently outperforms multiple baselines and ablation methods, underscoring its superiority. Importantly, we modify only the codebook training procedure without altering the quantization mechanism or downstream autoregressive model training. Thus, our work is complementary to ongoing advancements in these areas.

The remainder of this article is organized as follows. Section 2 presents related work. Section 4 details our proposed method. Experimental setup and results are discussed in Section 5. The paper is concluded in Section 6.

## 2. Related Work

### 2.1. Vector-Quantized Networks

Vector quantization networks (VQNs), initially proposed for image generation as VQVAE, map continuous embeddings to discrete codebook representations using vector quantization (VQ). The non-differentiable nature of VQ is addressed via straight-through estimators (STE, Bengio et al. (2013)). However, this presents the index collapse issue during codebook training, where only a small number of codes are active. To prevent the issue, Łańcucki et al. (2020); Zeghidour et al. (2021); Dhariwal et al. (2020) propose to reset unused codes periodically. In contrast, Kaiser et al. (2018) proposes

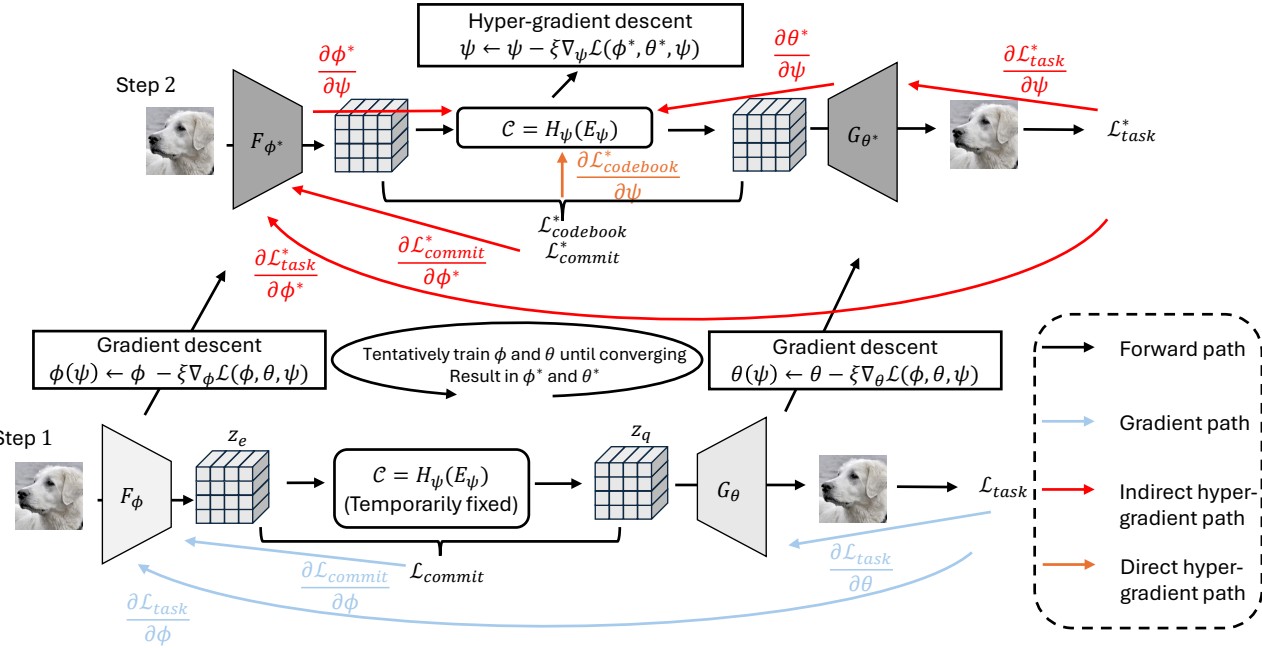

*Figure 2.* **Gradient paths in meta-quantization.** The lower level is optimized by gradient descent. The upper level is optimized by hyper-gradient descent with direct hyper-gradient and indirect hyper-gradient. Here, we use $\mathcal{L}^*_{\text{task}}$, $\mathcal{L}^*_{\text{commit}}$ and $\mathcal{L}^*_{\text{codebook}}$ to denote the corresponding loss computed with $\phi^*(\psi)$ and $\theta^*(\psi)$.

performing quantization in lower dimensionality to improve codebook utilization. Gumbel-VQ (Karpathy, 2021) enables differentiable quantization through continuous approximation of the $\arg\min$ operator. Other approaches include affine reparameterization (Huh et al., 2023), $l_2$ normalization (Yu et al., 2022), probabilistic formulations (Roy et al., 2018; Takida et al., 2022). More recently, VQGAN-LC (Zhu et al., 2024a) scales up the codebook size by initializing codebooks with features extracted by a pre-trained vision encoder, and focuses on training a projector that aligns the entire codebook with the feature distributions of the encoder. SimVQ (Zhu et al., 2024b) achieves even superior results by reparameterizing the code vectors through a linear transformation layer based on a learnable latent basis.

Other works, including FSQ (Mentzer et al., 2024) and LFQ (Yu et al., 2024), have explored using non-learnable, implicit codebooks. They replace the vector quantizer with scalar quantization schemes through rounding, making the codebook predefined and non-learnable. However, such methods suffer from reduced expressiveness due to the finite set of possible code values (specifically, a grid of integer values). Furthermore, FSQ and LFQ require representation to be projected into a reduced dimensionality, leading to significant information loss. These constraints motivate our focus on explicit codebook improvements. Our method shares certain similarities with them when viewing the codebook as a hyperparameter. FSQ and LFQ treat the codebook as a predefined hyperparameter; our work, on the other hand, searches for the optimal hyperparameter following the well-studied bi-level optimization-based hyperparameter optimization literature (Pedregosa, 2016; Finn et al., 2017).

### 2.2. Bi-level Optimization

Bi-level optimization (BLO) has found wide applicability in various machine learning tasks, with meta-learning (Finn et al., 2017; Rajeswaran et al., 2019) being one of its most notable applications. Other areas where BLO has been successfully employed include neural architecture search (NAS, Liu et al. (2019); Zhang et al. (2021)) and hyperparameter optimization (HPO, Lorraine et al. (2020); Franceschi et al. (2017)). Despite its broad usage, solving BLO problems remains challenging due to the inherently nested structure of the two optimization tasks. Gradient-based BLO (Choe et al., 2023) has garnered significant attention due to its scalability to high-dimensional problems with a large number of trainable parameters.

In this work, we extend the application of gradient-based BLO to propose a novel approach for codebook training within the vector quantization framework. Inspired by meta-learning, our MQ framework treats the codebook as hyperparameters, which are parameterized by a hyper-net. The meta-objective is to enhance the training of the encoder and decoder, replacing the previous VQ loss. The effectiveness of the codebook generated by the hyper-net is validated by assessing the performance of the encoder and decoder. This

process mirrors the practice of validating model initialization in meta-learning. A similar strategy involving hyper-gradient descent with inner loop unrolling is employed in our work, a technique that is also commonly found in the meta-learning literature.

## 3. Preliminary

A vector-quantized network (VQN) is a neural network that incorporates a vector-quantization (VQ) layer $h_{\mathcal{C}}(\cdot)$, as described by the following equation:

$$\mathbf{y} = G_\theta(h_{\mathcal{C}}(F_\phi(\mathbf{x}))) = G_\theta(h_{\mathcal{C}}(\mathbf{z}_e)) = G_\theta(\mathbf{z}_q) \quad (1)$$

Here, $\mathbf{z}_e$ denotes the embedding obtained by applying the encoder $F_\phi$ (parameterized by $\phi$) to the input $\mathbf{x}$. $\mathbf{z}_q$ represents the quantized embedding, which is obtained by applying the VQ layer $h_{\mathcal{C}}$ to $\mathbf{z}_e$. The output $\mathbf{y}$ is generated by the decoder $G_\theta$ (parameterized by $\theta$), which takes $\mathbf{z}_q$ as input.

The VQ layer $h_{\mathcal{C}}(\cdot)$ performs quantization on $\mathbf{z}_e$ by selecting a vector from the codebook $\mathcal{C} = \{\mathbf{e}_1, \mathbf{e}_2, \ldots, \mathbf{e}_K\}$ based on a distance measure $d(\cdot, \cdot)$,

$$\mathbf{z}_q = \mathbf{e}_k, \quad \text{where} \quad k = \arg\min_j d(\mathbf{z}_e, \mathbf{e}_j) \quad (2)$$

Here, a learned vector $\mathbf{e}_i$ is referred to as the code, and the index $i$ denotes the corresponding code. The Euclidean distance is commonly used as the distance measure for $d(\cdot, \cdot)$ (van den Oord et al., 2017). The quantized embedding $\mathbf{z}_q$ is a subset of $\mathcal{C}$, and updating $\mathbf{z}_q$ corresponds to partially updating $\mathcal{C}$.

The task loss $\mathcal{L}_{\text{task}}(G_\theta(h_{\mathcal{C}}(F_\phi(\mathbf{x}))), \mathbf{y})$ is not continuously differentiable due to the $\arg\min$ operator in $h_{\mathcal{C}}$. To address this issue, a straight-through estimator (STE, Bengio et al. (2013)) is applied, where the non-differentiable part $\frac{\partial \mathbf{z}_q}{\partial \mathbf{z}_e}$ is ignored:

$$\frac{\partial \mathcal{L}_{\text{task}}}{\partial \phi} \approx \frac{\partial \mathcal{L}_{\text{task}}}{\partial \mathbf{y}} \frac{\partial \mathbf{y}}{\partial \mathbf{z}_q} \frac{\partial \mathbf{z}_e}{\partial \phi} \quad (3)$$

To ensure an accurate STE, $\mathbf{z}_e$ and $\mathbf{z}_q$ are aligned using two additional losses:

$$\mathcal{L}_{\text{commit}}(\mathbf{z}_q, \mathbf{z}_e) = d(\mathbf{z}_e, \text{sg}[\mathbf{z}_q]) \quad (4)$$
$$\mathcal{L}_{\text{codebook}}(\mathbf{z}_q, \mathbf{z}_e) = d(\text{sg}[\mathbf{z}_e], \mathbf{z}_q) \quad (5)$$

Here, sg denotes the stop-gradient operator, which treats the entire term as a constant with zero partial derivatives. The commitment loss $\mathcal{L}_{\text{commit}}$ encourages the embedding to move toward the selected codes, whereas the codebook loss $\mathcal{L}_{\text{codebook}}$, also known as the vector quantization objective, pushes the selected codes toward the centroids of the embedding.

Overall, a differentiable objective is minimized:

$$\min_{\phi, \theta, \mathcal{C}} \mathbb{E}_{(\mathbf{x}, \mathbf{y}) \sim \mathcal{D}} [\mathcal{L}_{\text{task}}(G_\theta(h_{\mathcal{C}}(F_\phi(\mathbf{x}))), \mathbf{y})$$
$$+ \alpha \cdot \mathcal{L}_{\text{commit}}(h_{\mathcal{C}}(F_\phi(\mathbf{x})), F_\phi(\mathbf{x})) \quad (6)$$
$$+ \beta \cdot \mathcal{L}_{\text{codebook}}(h_{\mathcal{C}}(F_\phi(\mathbf{x})), F_\phi(\mathbf{x}))] \quad (7)$$

where $\alpha$ and $\beta$ are scalars that balance the loss combination. In this training framework, the decoder optimizes the first term, the encoder optimizes both the first and the middle terms, and the codebook is optimized only by the last term.

## 4. Methodology

### 4.1. Hyper-Net Reparameterization

We avoid directly optimizing the codebook by introducing a hyper-net $H_\psi$, which takes as input a learnable embedding $E_\psi$ and predicts the codes in the codebook, which are then used for vector quantization. More formally, this can be expressed as $\mathcal{C} = H_\psi(E_\psi)$. As shown on the right side of Figure 1, the trainable parameters (denoted by $\psi$) in the quantization layer are the hyper-net and the learnable embedding rather than the codebook itself. For notation simplicity, we use $H_\psi$ to denote $H_\psi(E_\psi)$ in the remaining parts of this paper.

Once the first stage of image modeling is completed, the hyper-net can be discarded, retaining only the generated codebook for downstream tasks. This codebook acts as a direct replacement for the codebook used in previous training procedures. Furthermore, since the generated codebook is a complicated non-linear combination of the learnable embedding through the mapping of the hyper-net, even if only a few codes are selected, a substantial part of the hyper-net still receives gradients. This allows the hyper-net to generate a codebook that better aligns with the embedding distribution.

### 4.2. A Bi-level Optimization Framework

In the spirit of meta-learning, we propose to optimize the hyper-net and encoder-decoder pair by solving a bi-level optimization problem. Specifically, we optimize the hyper-net $H_\psi$ at the upper level and the encoder $F_\phi$ and decoder $G_\theta$ at the lower level. In both levels, we consider a loss $\mathcal{L}$ defined in the same form as in Equation (7), i.e., the sum of $\mathcal{L}_{\text{task}}$, $\mathcal{L}_{\text{commit}}$, and $\mathcal{L}_{\text{codebook}}$.

**Lower Level** At the lower level, we train the encoder $F_\phi$ and decoder $G_\theta$ by minimizing $\mathcal{L}(\phi, \theta, \psi)$. Specifically, we aim to find the optimal values of $\phi$ and $\theta$ with $\psi$ temporarily fixed, resulting in the following optimization problem:

$$\phi^*(\psi), \theta^*(\psi) = \arg\min_{\phi, \theta} \mathcal{L}(\phi, \theta, \psi) \quad (8)$$

Here, $\phi^*(\psi)$ and $\theta^*(\psi)$ denote the optimal solutions for $\phi$ and $\theta$, which are functions of $\psi$, since the lower-level problem does not take $\psi$ as an argument.

**Upper Level** At the upper level, the hyper-net $H_\psi$ is trained by minimizing the loss of the same functional form but using $\phi^*(\psi)$ and $\theta^*(\psi)$ that were optimally learned at the lower level as arguments. The loss then only depends on $\psi$, and the upper-level optimization problem is formulated as:

$$\min_\psi \mathcal{L}(\phi^*(\psi), \theta^*(\psi), \psi) \tag{9}$$

**A Bi-level Optimization Framework** By integrating the two levels of optimization problems, we present the overall bi-level optimization problem as:

$$\min_\psi \mathcal{L}(\phi^*(\psi), \theta^*(\psi), \psi)$$
$$s.t. \quad \phi^*(\psi), \theta^*(\psi) = \arg\min_{\phi,\theta} \mathcal{L}(\phi, \theta, \psi) \tag{10}$$

The two levels of optimization problems are mutually dependent on each other. The solution to the optimization problem at the lower level, $\phi^*(\psi)$ and $\theta^*(\psi)$ serves as a parameter for the upper-level problem, while the non-optimal variable $\psi$ at the upper level acts as a parameter for the lower-level problem. By solving the two interconnected problems jointly, we can learn $\phi^*$, $\theta^*$, and $\psi^*$ in an end-to-end manner.

---

**Algorithm 1** Meta-Quantization

---

**Require:** Dataset $\mathcal{D}$
1: Initialize $\phi$, $\theta$ and $\psi$ for $F_\phi$, $G_\theta$, $H_\psi$ and $E_\psi$.
2: **while** not converged **do**
3:     Update $\psi$ by gradient descent as $\nabla_\psi \mathcal{L}(\phi - \xi\nabla_\phi\mathcal{L}(\phi,\theta,\psi), \theta - \xi\nabla_\theta\mathcal{L}(\phi,\theta,\psi), \psi)$
4:     ($\xi = 0$ if using alternated optimization)
5:     Update $\phi$ and $\theta$ by gradient descent as $\nabla_\phi\mathcal{L}(\phi,\theta,\psi)$ and $\nabla_\theta\mathcal{L}(\phi,\theta,\psi)$
6: **end while**
**Ensure:** $\phi^*$, $\theta^*$ and $\psi^*$

---

**Optimization Algorithm** We employ an efficient gradient-based optimization algorithm to solve the bi-level optimization problem presented in Equation (10), with the two levels optimized iteratively until convergence. Related convergence analyses of this type of gradient-based bi-level optimization algorithm can be found in Pedregosa (2016), Rajeswaran et al. (2019), and references therein.

Gradient descent can be applied directly to the lower-level problem; however, a significant challenge arises at the upper level: computing the hyper-gradient, i.e., the gradient of the upper-level objective with respect to $\psi$, can be computationally prohibitive due to the lack of an analytical solution

for $\phi^*(\psi)$ and $\theta^*(\psi)$. To address this, we adopt a one-step approximation (Finn et al., 2017):

$$\nabla_\psi \mathcal{L}(\phi^*(\psi), \theta^*(\psi), \psi)$$
$$\approx \nabla_\psi \mathcal{L}(\phi - \xi\nabla_\phi\mathcal{L}(\phi,\theta,\psi), \theta - \xi\nabla_\theta\mathcal{L}(\phi,\theta,\psi), \psi) \tag{11}$$

where $\xi$ is the learning rate for the lower-level problem. One-step unrolled approximated solutions, $\phi'(\psi) = \phi - \xi\nabla_\phi\mathcal{L}(\phi,\theta,\psi)$ and $\theta'(\psi) = \theta - \xi\nabla_\theta\mathcal{L}(\phi,\theta,\psi)$, are used as surrogates for $\phi^*(\psi)$ and $\theta^*(\psi)$. This is equivalent to introducing a surrogate objective $\mathcal{L}(\phi - \xi\nabla_\phi\mathcal{L}(\phi,\theta,\psi), \theta - \xi\nabla_\theta\mathcal{L}(\phi,\theta,\psi), \psi)$ that closely resembles the upper-level objective in Equation (9).

Existing meta-learning methods compute Equation (11) by either backpropagating through the optimization process at the lower level (Finn et al., 2017) or applying implicit differentiation with a Hessian matrix of the inner optimization problem (Rajeswaran et al., 2019). However, as the problem size scales, the memory and computational burden grow significantly. Therefore, we employ a further approximation by noticing that Equation (11) can be computed using the chain rule, followed by a finite difference approximation (Liu et al., 2019) as:

$$\nabla_\psi \mathcal{L}(\phi - \xi\nabla_\phi\mathcal{L}(\phi,\theta,\psi), \theta - \xi\nabla_\theta\mathcal{L}(\phi,\theta,\psi), \psi) \tag{12}$$
$$= \nabla_\psi \mathcal{L}(\phi', \theta', \psi) \tag{13}$$
$$- \xi\nabla^2_{\psi,\phi}\mathcal{L}(\phi,\theta,\psi)\nabla_{\phi'}\mathcal{L}(\phi',\theta',\psi)$$
$$- \xi\nabla^2_{\psi,\theta}\mathcal{L}(\phi,\theta,\psi)\nabla_{\theta'}\mathcal{L}(\phi',\theta',\psi) \tag{14}$$
$$\approx \nabla_\psi \mathcal{L}(\phi', \theta', \psi) \tag{15}$$
$$- \xi\frac{\nabla_\psi\mathcal{L}(\phi^+,\theta,\psi) - \nabla_\psi\mathcal{L}(\phi^-,\theta,\psi)}{2\epsilon}$$
$$- \xi\frac{\nabla_\psi\mathcal{L}(\phi,\theta^+,\psi) - \nabla_\psi\mathcal{L}(\phi,\theta^-,\psi)}{2\epsilon} \tag{16}$$

where $\phi^\pm = \phi \pm \epsilon\nabla_{\phi'}\mathcal{L}(\phi',\theta',\psi)$, $\theta^\pm = \theta \pm \epsilon\nabla_{\theta'}\mathcal{L}(\phi',\theta',\psi)$, and $\epsilon$ is a small scalar. The finite difference is applied to approximate the matrix-vector multiplication term in Equation (14) for efficient computation.

### 4.3. Gradient Analysis

Essentially, meta-quantization introduces a meta-objective $\mathcal{L}(\phi^*(\psi), \theta^*(\psi), \psi)$ in place of the VQ objective for the quantizer training. While it shares the same functional form as the previous framework (Equation (7)), the arguments $\phi$ and $\theta$ are set to their optimal values $\phi^*(\psi)$ and $\theta^*(\psi)$. We demonstrate that this meta-objective improves the gradient guidance for $\psi$ by performing a gradient analysis on the one-step-unrolled surrogate loss using the chain rule.

Define $\mathcal{L}'(\phi, \theta, \psi) = \mathcal{L}(\phi'(\psi), \theta'(\psi), \psi) = \mathcal{L}(\phi - $

$\xi\nabla_\phi\mathcal{L}(\phi,\theta,\psi), \theta-\xi\nabla_\theta\mathcal{L}(\phi,\theta,\psi),\psi)$. We then have

$$\frac{d\mathcal{L}'}{d\psi} = \frac{\partial\mathcal{L}'}{\partial\psi} + \frac{\partial\phi'}{\partial\psi}\times\frac{\partial\mathcal{L}'}{\partial\phi'} + \frac{\partial\theta'}{\partial\psi}\times\frac{\partial\mathcal{L}'}{\partial\theta'} \qquad (17)$$

The last two terms on the right-hand side, especially $\frac{\partial\phi'}{\partial\psi}$ and $\frac{\partial\theta'}{\partial\psi}$, referred to as the best-response Jacobian in the literature (Choe et al., 2023), *capture how the encoder-decoder pair reacts to changes of the quantizer*. Therefore, the update of $\psi$ must consider not only the direct gradient from the loss ($\frac{\partial\mathcal{L}'}{\partial\psi}$) for minimizing quantization error but also additional information about indirect gradients—how the encoder and decoder would respond to changes in the quantizer ($\frac{\partial\phi'}{\partial\psi}$ and $\frac{\partial\theta'}{\partial\psi}$), and their performance potential ($\frac{\partial\mathcal{L}'}{\partial\phi'}$ and $\frac{\partial\mathcal{L}'}{\partial\theta'}$). The encoder and decoder select their best response by conducting gradient descent, which the quantizer accounts for. This facilitates the finding of a globally optimal quantizer, thereby improving its stability and robustness. See also Figure 2 for the gradient path and hyper-gradient path used in the lower and upper levels, respectively. Additionally, we observe that by employing this strategy, $\psi$ can now receive a gradient from $\mathcal{L}_{\text{task}}$. For example, the first terms of $\mathcal{L}'$, i.e., $\mathcal{L}'_{\text{task}}$, depend on $\phi'$, which in turn depends on $\psi$. This joint effort enables the $\mathcal{L}_{\text{task}}$ to influence $\psi$ during backpropagation.

## 5. Experiment

In this section, we first evaluate MQ on image generative modeling, including image reconstruction and image generation, using the VQVAE architecture (van den Oord et al., 2017). The experiments are done on small-scale datasets: CIFAR10 (Krizhevsky et al., 2009) and CelebA (Liu et al., 2015). We then scale up to a larger experimental setting on FFHQ (Karras et al., 2019) and ImageNet (Deng et al., 2009) with VQGAN (Esser et al., 2021), which involves perceptual loss and adversarial loss as task losses. We refer to the resulting methods combined with MQ "MQVAE" and "MQGAN", respectively. For the bi-level optimization algorithm implementation, our code is mainly based on the Betty library (Choe et al., 2023). We release our code at GitHub [1] for future research.

### 5.1. Evaluation with VQVAE

**Setup** We first show preliminary experimental results with the VQVAE architecture on CIFAR10 (Krizhevsky et al., 2009) at $32 \times 32$ resolution and CelebA (Liu et al., 2015) at $128 \times 128$ resolution. For CelebA, images undergo random cropping to $140 \times 140$ pixels, followed by resizing the smaller dimension to 128 while maintaining aspect ratio. No additional augmentations are used for CIFAR10.

The backbone architecture of the autoencoder for all com-

---

[1]https://github.com/t2ance/MQVAE

pared methods follows Takida et al. (2022) with 64 channels. A codebook size of 1024 is used across all methods. For the hyper-net configuration, an MLP is used. It first lifts the 32-dimensional learned embedding to 256 dimensions, then projects it back to 32 dimensions to form the codebook entries for quantization, using Tanh as the activation function.

Models are trained using Mean Squared Error (MSE) as the reconstruction loss; no perceptual or discriminative losses are used. We employ the Adam optimizer (Kingma & Ba, 2015) with momentum set to $(0.9, 0.95)$ and an initial learning rate of 1e-4. The learning rate follows a linear warmup for the first $10\%$ of epochs, then a half-cycle cosine decay. No weight decay is applied to the quantizer. Training runs for a maximum of 90 epochs, with early stopping if performance saturates.

Evaluation metrics include MSE (the same as training loss), LPIPS (Zhang et al., 2018), and model perplexity. Perplexity is defined as $2^{H(p)}$, where $H(p)$ is the entropy over the codebook likelihood; a higher value is preferred as it implies more uniform code usage.

Our method is compared against VQVAE (van den Oord et al., 2017), SQVAE (Takida et al., 2022), and Gumbel-VQVAE (Karpathy, 2021; Esser et al., 2021). Comparisons also include VQ training techniques such as $l_2$ normalization (Yu et al., 2022), least-recently-used (LRU) replacement (Łańcucki et al., 2020), affine reparameterization, and synchronized training (OPT) (Huh et al., 2023). For other unspecified configurations, experimental settings from Huh et al. (2023) are adopted, and their results are cited for comparison.

**Image Reconstruction** Table 1 shows that almost all baselines significantly improve training stability and the reconstruction performance of generative models. However, as analyzed in the previous section, existing methods suffer from the limitations that lack a direct gradient flow from task loss to codebook, resulting in degraded reconstruction performance. Notably, our method provides the best improvements, particularly with respect to the MSE, which is the task loss. This verifies the effectiveness of our method for achieving superior image quality and codebook utilization.

**Image Generation** We extend our results to the image generation task using MaskGIT (Chang et al., 2022) on CelebA, utilizing the codebook trained in the first stage directly. A slimmed-down version of MaskGIT (Huh et al., 2023) is used: VQGAN using 32 channels instead of 128 and transformer using 8 blocks instead of 24. In Table 2, the generation results improve from the baseline by 90.4 FID and 74.8 FID from the best-performing variation. In

*Table 1.* Comparison between various methods on image reconstruction task.

| | Method | MSE $(10^{-3})$ | Perplexity | LPIPS |
|---|---|---|---|---|
| CIFAR10 | VQVAE (van den Oord et al., 2017) | 5.65 | 14.0 | 0.54 |
| | $l_2$ (Yu et al., 2022) | 3.21 | 57.0 | 0.36 |
| | LRU (Łańcucki et al., 2020) | 4.07 | 109.8 | 0.43 |
| | $l_2$ + LRU | 3.24 | 115.6 | 0.35 |
| | SQVAE (Takida et al., 2022) | 3.36 | 769.3 | 0.39 |
| | Gumbel-VQVAE (Karpathy, 2021) | 6.16 | 20.3 | 0.57 |
| | Affine (Huh et al., 2023) | 5.15 | 69.5 | 0.51 |
| | OPT (Huh et al., 2023) | 4.73 | 15.5 | 0.48 |
| | Affine + OPT (Huh et al., 2023) | 4.00 | 79.3 | 0.43 |
| | MQVAE (ours) | **3.05** | **783.4** | **0.29** |
| CELEBA | VQVAE(van den Oord et al., 2017) | 10.02 | 16.2 | 0.27 |
| | $l_2$ (Yu et al., 2022) | 6.49 | 188.7 | 0.18 |
| | LRU (Łańcucki et al., 2020) | 4.77 | 676.4 | 0.16 |
| | $l_2$ + LRU | 4.93 | 861.7 | 0.15 |
| | SQVAE (Takida et al., 2022) | 9.17 | 769.1 | 0.27 |
| | Gumbel-VQVAE (Karpathy, 2021) | 7.34 | 96.7 | 0.23 |
| | Affine(Huh et al., 2023) | 7.47 | 112.6 | 0.22 |
| | OPT(Huh et al., 2023) | 7.78 | 30.5 | 0.23 |
| | Affine + OPT (Huh et al., 2023) | 6.60 | 186.6 | 0.18 |
| | MQVAE (Ours) | **3.10** | **877.8** | **0.14** |

contrast, our method improves the FID to $70.5$, achieving the best performance among all baselines.

*Table 2.* Image generation on CelebA using MaskGIT.

| Method | FID |
|---|---|
| MaskGIT (Chang et al., 2022) | 90.4 |
| $l_2$ (Yu et al., 2022) | 81.5 |
| LRU (Łańcucki et al., 2020) | 79.7 |
| Affine + OPT (Huh et al., 2023) | 74.8 |
| MQVAE (Ours) | **70.5** |

### 5.2. Evaluation with VQGAN

**Setup** In this subsection, we evaluate the scalability of our method using the VQGAN (Esser et al., 2021) architecture, with all compared methods utilizing its original encoder and decoder. These backbone architecture choices directly follow Esser et al. (2021) to ensure comparable results. Experiments were conducted on the ImageNet-1K (Deng et al., 2009) and FFHQ (Karras et al., 2019) datasets. For ImageNet-1K, input images were processed at $128 \times 128$ pixels. This involved resizing the image's smaller dimension to $128$ while maintaining aspect ratio, followed by a $128 \times 128$ random crop and a $50\%$ probability horizontal flip. FFHQ images were processed at $256 \times 256$ pixels directly. The U-Net-based encoder (Ronneberger et al., 2015) downsamples input images to a $16 \times 16$ feature map for each case. The quantizer then converts this into a token

*Table 3.* Reconstruction performance on FFHQ.

| Method | Utilization (%) | rFID | LPIPS |
|---|---|---|---|
| VQGAN† (Esser et al., 2021) | 2.3 | 5.25 | 0.12 |
| VQGAN-FC† (Yu et al., 2022) | 10.9 | 4.86 | 0.11 |
| VQGAN-EMA† (Razavi et al., 2019) | 68.2 | 4.79 | 0.10 |
| VQGAN-LC (Zhu et al., 2024a) | 99.9 | 4.65 | 0.10 |
| MQGAN (Ours) | **100.0** | **4.25** | **0.08** |

*Table 4.* Reconstruction performance on ImageNet-1k with a resolution of $128 \times 128$.

| Method | Utilization (%) | rFID | LPIPS |
|---|---|---|---|
| VQGAN (Esser et al., 2021) | 1.4 | 3.74 | 0.17 |
| VQGAN-EMA (Razavi et al., 2019) | 4.5 | 3.23 | 0.15 |
| VQGAN-FC (Yu et al., 2022) | **100.0** | 2.63 | 0.13 |
| FSQ (Mentzer et al., 2024) | **100.0** | 2.80 | 0.13 |
| LFQ (Yu et al., 2024) | **100.0** | 2.88 | 0.13 |
| VQGAN-LC (Zhu et al., 2024a) | **100.0** | 2.40 | 0.13 |
| SimVQ (Zhu et al., 2024b) | **100.0** | 2.24 | 0.13 |
| MQGAN (Ours) | **100.0** | **2.13** | **0.12** |

*Table 5.* Image generation on FFHQ.

| Method | Utilization (%) | FID |
|---|---|---|
| VQGAN-FC† (Yu et al., 2022) | 10.9 | 3.23 |
| VQGAN-EMA† (Razavi et al., 2019) | 68.2 | 4.87 |
| VQGAN-LC (Zhu et al., 2024a) | **99.9** | 3.05 |
| MQGAN (Ours) | **99.9** | **2.87** |

map of the same dimensions (256 tokens), which the U-Net-based decoder uses for image reconstruction. For the hyper-net configuration, an MLP is used. It first lifts the 32-dimensional learned embedding to 256 dimensions, then projects it back to 32 dimensions to form the codebook entries for quantization, using Tanh as the activation function. The number of codes is set to $65,536$ for ImageNet and $16,384$ for FFHQ, consistent with comparison methods. For initialization, codebook embeddings are initialized using a simple Gaussian distribution, avoiding the need for a pretrained model as in Zhu et al. (2024a). The hyper-net transformation uses PyTorch's default initialization.

We train for 20 epochs on ImageNet-1K and $800$ epochs on FFHQ, employing early stopping if performance saturates. The Adam optimizer (Kingma & Ba, 2015) with optimizer momentum of $(0.5, 0.9)$ is used with an initial learning rate of 1e-4. The learning rate undergoes a linear warmup for the first $10\%$ of epochs, followed by a half-cycle cosine decay schedule. Evaluation involves several metrics. Reconstruction FID is measured using the GAN-trained autoencoder with validation images passed through the quantized autoencoder; this reflects the FID a Stage II transformer could achieve with perfect data modeling. Codebook usage is determined by the fraction of codewords utilized at least

once when encoding the validation set, following (Mentzer et al., 2024) and (Zhu et al., 2024b). Additionally, generation FID is reported for the second stage involving a trained transformer, obtained by decoding representations sampled (potentially class-conditionally) with the transformer.

Our method is compared against several baselines: standard VQGAN (Esser et al., 2021), VQGAN-FC (Yu et al., 2022), VQGAN-EMA (Razavi et al., 2019), VQGAN-LC (Zhu et al., 2024a), SimVQ (Zhu et al., 2024b), and implicit codebook methods FSQ (Mentzer et al., 2024) and LFQ (Yu et al., 2024).

**Image Reconstruction**   In the image reconstruction task, we evaluate performance using rFID, LPIPS, PSNR, and SSIM metrics on the validation sets of ImageNet and FFHQ. Table 3 and Table 4 present the reconstruction performance for FFHQ and ImageNet, respectively. For FFHQ, results cited from Zhu et al. (2024a) are denoted by †. For ImageNet, all results of baselines are cited from Zhu et al. (2024b).

We make the key observation that our method consistently achieves the best performance with a codebook utilization rate of over 99% on both datasets. When compared with implicit codebooks such as FSQ (Mentzer et al., 2024) and LFQ (Yu et al., 2024) in Table 4, MQGAN also achieves superior performance. This can be attributed to the reduction of expressiveness and model capacity in FSQ and LFQ, while our method follows a learnable, explicit codebook approach, leading to better performance when the codebook utilization rate is high.

**Image Generation**   We follow VQGAN, using a causal Transformer decoder (Radford et al., 2019) with 24 layers, 16 heads per attention layer, and a latent dimension of 1024. For FFHQ, the FID score is determined using 50K generated images in comparison with the combined training and validation sets of FFHQ. Table 5 displays the unconditional generation results on the FFHQ dataset. Compared with baselines, the generative model GPT demonstrates improved performance and codebook utilization with the integration of our MQGAN.

### 5.3. Ablation Studies

*Table 6.* Ablation study on bi-level optimization (BLO).

|  | Perplexity | MSE | PSNR | SSIM |
|---|---|---|---|---|
| w/o BLO | 836.4 | 3.65 | 28.3 | 77.8 |
| w/ BLO ($\xi = 0$) | 851.4 | 3.61 | 28.8 | 80.3 |
| w/ BLO | **877.8** | **3.10** | **29.4** | **82.1** |

Unless otherwise specified, we evaluate reconstruction performance in terms of PSNR and SSIM (image quality), and

*Table 7.* Ablation study on hyper-net.

| Hyper-net type | Perplexity | MSE | PSNR | SSIM |
|---|---|---|---|---|
| Identity | 9.3 | 5.54 | 25.7 | 70.5 |
| Linear-1 | 334.28 | 3.50 | 28.4 | 78.6 |
| Linear-2 | 52.8 | 4.25 | 27.2 | 75.2 |
| MLP | **877.8** | **3.10** | **29.4** | **82.1** |

MSE (task loss) on CelebA across all studies.

**Effectiveness of Bi-level Optimization**   In Equation (14), we see that $\xi$ controls the influence of task-specific objectives. We set the value of $\xi$ to zero to investigate the effectiveness of indirect gradient, denoted by "w/ BLO ($\xi = 0$)". We also provide the results of "w/o BLO", which denotes using hyper-net reparameterization without bi-level optimization in MQ.

From Table 6, we draw the observation that using a meta-learning-inspired objective significantly improves image modeling ability. This improvement validates the core idea of our approach to directly incorporate task loss into the codebook training via the meta-objective.

**Type of Hyper-Net**   We evaluate the effectiveness of the hyper-net by varying its specific parameterization. We compare the following types:

1. **Identity:** A simple baseline without hyper-net.

2. **Linear-1:** A linear layer that projects an embedding $E \in \mathbb{R}^{K \times d'}$ to $\mathcal{C} \in \mathbb{R}^{K \times d}$, i.e., each code is projected to a different dimensionality.

3. **Linear-2:** A linear layer that projects an embedding $E \in \mathbb{R}^{K' \times d}$ to $\mathcal{C} \in \mathbb{R}^{K \times d}$, i.e., each code is a learned linear combination of the basis $E$.

4. **MLP:** An MLP with one hidden layer which first projects an embedding $E \in \mathbb{R}^{K \times d'}$ to hidden neurons of $\mathbb{R}^{K \times d''}$, and then to $\mathcal{C} \in \mathbb{R}^{K \times d}$, constituting a universal approximator to almost any continuous functions.

For all ablative methods, we set $d = d' = 32$ and $K = K' = 1024$. Table 7 shows that employing a more complex hyper-net consistently leads to better performance. This is attributed to the enhanced meta-learner, resulting in better generative modeling performance. Therefore, we employ an MLP reparameterization in all our experiments.

### 5.4. Visualization of Embedding and Code Distributions

Figure 3 shows the distribution of the extracted features and codes for two models: VQGAN and our MQGAN.

The visualization is created using t-SNE (Van der Maaten & Hinton, 2008). The codebook entries are highlighted in blue, while the extracted features are shown in red. A larger amount of overlaying denotes a higher utilization rate. Results show that MQGAN tends to have a larger overlap between two distributions than naive VQGAN, and is able to enhance code utilization.

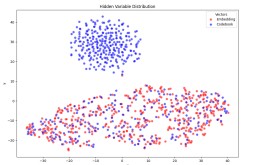 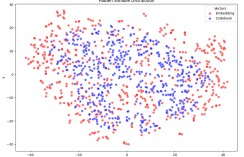

(a) Code distribution of VQ    (b) Code distribution of MQ

*Figure 3.* Comparison between code distribution. Notably, MQ has a larger overlap area between code and extracted embedding, resulting in a higher code utilization rate.

### 5.5. Visualization between Direct and Indirect Gradients

The hyper-gradient consists of two parts- the direct and indirect gradients- and it is worth seeing whether both of them play an important role. We visualize the norm of direct and indirect gradients of the hyper-gradient during a single run. A similar pattern can be observed in all the experiments. As shown in Figure 4, both of the norms are in a magnitude of 1e-2. Additionally, the cosine similarity remains close to zero during training, therefore, neither of them should be excluded from training.

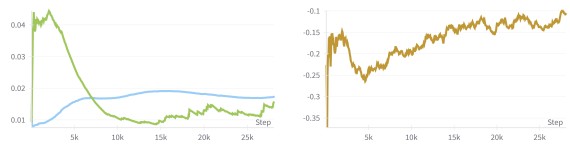

*Figure 4.* Visualization between direct and indirect gradients. (a) Comparison in terms of gradient norm. (b) Cosine similarity between the two gradients.

### 5.6. Qualitative Evaluation

In Figure 5, we present the reconstruction results at a resolution of $256 \times 256$ for our MQGAN. The introduction of diversified gradient paths and a large-scale codebook facilitates the generation of images, as revealed by the qualitative results.

### 5.7. Training Cost Comparison

We conducted additional experiments to show the efficiency of MQ training. We compare VQVAE and MQVAE in terms

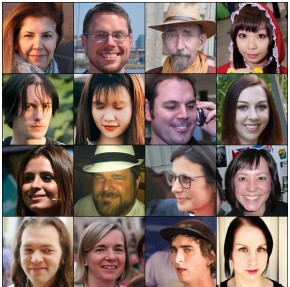 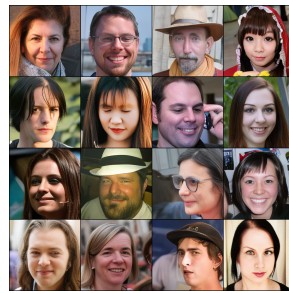

(a) Samples from dataset    (b) Samples generated

*Figure 5.* Qualitative evaluation of reconstruction

of time and memory cost. When evaluated on the CelebA dataset with a batch size of $128$, the increase in memory usage is marginal and acceptable in practice. For time comparison, we set VQVAE as the baseline, which needs around $3.6$ hours to finish the training of $50k$ steps (and does not improve after that). In Table 8, we observe that MQVAE only requires $2.4$ hours to reach the same LPIPS score as VQVAE (around $9.5k$ steps), and can keep improving after that. This demonstrates that our method converges much faster than VQVAE and is able to outperform baselines with extended training time.

*Table 8.* Comparison between various methods on time and memory cost.

| Method | Memory (GB) | Wall time to reach VQVAE (h) | Total wall time (h) |
|---|---|---|---|
| VQVAE | 7.29 | 3.6 | 3.6 |
| MQVAE | 7.35 | 2.4 | 12.2 |

## 6. Conclusion

We propose meta-quantization (MQ), a novel bi-level optimization-based vector-quantization framework inspired by meta-learning. Building upon prior VQ mechanisms, our method trains the quantizer and the autoencoder at different levels within a bi-level optimization problem. Furthermore, we reparameterize the codebook using a hyper-net, which generates the codebook from a learnable embedding and updates it holistically. In essence, our approach replaces the VQ objective of the codebook with a meta-objective, which aims to optimize the learning potential of the autoencoder. This facilitates a gradient flow from the task loss to the quantizer, thereby improving overall performance. Empirical studies across various experimental settings demonstrate that MQ outperforms the prior VQ method and its variants, underscoring its effectiveness.

## Impact Statement

This paper presents work whose goal is to advance the field of Machine Learning. Specifically, our work aims to improve existing discrete representation learning methods with potential applications for a wide range of tasks, such as image generation. There are many potential societal consequences of our work, none of which we feel must be specifically highlighted here.

## Acknowledgment

This work was supported in part by the National Natural Science Foundation of China (Grant No. 62276121), the TianYuan funds for Mathematics of the National Science Foundation of China (Grant No. 12326604), and Shenzhen International Research Cooperation Project (Grant No. GJHZ20220913142611021).

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
