# OpenReview forum: "Learning to Quantize for Training Vector-Quantized Networks"
_ICML.cc/2025/Conference — ICML 2025 poster_

### Official Review · Reviewer_VZz4 · 2025-03-14

**Overall Recommendation:** 3

**Summary:**

This paper proposes an improvement to the STE method for training VQ networks. While the backpropagated gradient bypasses the codebook in the STE framework, this paper proposes Meta Quantization (MQ), which adopts a bi-level optimization strategy and learn quantization with a hyper-net in a meta-learning fashion. This enables the task loss to reach the codebook through multiple routes. It can also discard the hyper-net and retain only codebook weights, without affecting downstream tasks. Empirical evidence suggests better reconstruction and generation quality across various datasets.

**Claims And Evidence:**

•	Claim: The paper argues that introducing MSE loss for codebook optimizing by hyper-net could enhance the training.
o	Question: Although the experiments have shown the effectiveness of the generated codebook by hyper-net on image reconstruction and generation. There is little direct discussion of computational cost or difficulty of converging using the meta-learning strategy.
o	Potential Improvement: It will be helpful to show the performance between a conventional VQN and MQ trained with the same epochs, or figure out the steps needed until they have converged, respectively.

•	Claim: Hyper-net reparameterization and meta-learning help avoid code collapse.
o	Question: The authors show near-100% code usage on image reconstruction and generation, however, it is not clear about the process of codebook optimization itself.
o	Potential Improvement: It will be better to demonstrate how codes are generated, selected, and which part is optimized. As mentioned in Sec. 4.1 (only a few codes are selected, a substantial part of the hyper-net still receives gradients).

**Essential References Not Discussed:**

The paper does cite and discuss relevant references, including meta-learning strategies and VQNs that focus on codebook collapse. However, it is better to expand the discussion of recent VQN researches dealing with the collapse and compare with them, such as CVQ-VAE [online’ 23] by zheng et al. published in ICCV 2023.

**Experimental Designs Or Analyses:**

Design Strength: The authors compare their method to a variety of widely used VQN baselines on multiple tasks. They measure codebook usage, reconstruction quality, and generation quality, providing a holistic overview.

Potential Weakness:
1.	For reconstruction evaluation in Tab. 1, it follows [Straightening’ 23] by Huh et al. published in PMLR 2023, but some results are missed. E.g. VQVAE+Affine+OPT+replace+l_2 shows an MSE of 1.74, and LPIPS of 0.227, better than MQVAE proposed in this paper with MSE of 3.05 and LPIPS of 0.29. It’s better to involve these missing results and analyze about the gap.
2.	Recent research such as CVQ-VAE [online’ 23] by zheng et al. published in ICCV 2023 should also be considered for more comprehensive comparison.

**Methods And Evaluation Criteria:**

•	Methods: The authors optimize the codebook with a hyper-net and introduce task loss for explicit codebook improvements. This makes sense for index collapse problem and potential better image reconstruction or generation performance.
•	Evaluation Criteria: The paper relies on standard generative modeling metrics such as MSE, LPIPS, SSIM, and FID across established datasets, following [Straightening’ 23] by Huh et al. published in PMLR 2023.
•	Potential Weakness: Since the method focus on explicit codebook improvements, it will be better to illustrate the codebook distribution, using tools such as tSNE, to directly validate the effectiveness of the proposed method.

**Other Comments Or Suggestions:**

•	Implementation Details: Additional clarifications about hyperparameter tuning and computational cost might help replicate the strong results.
•	Downstream Utility: While the paper shows standard metrics, some real-world usage scenarios or domain-specific tasks (e.g., image segmentation, speech coding) might illustrate practical advantages.

**Other Strengths And Weaknesses:**

Strengths:
1.	This paper proposes a novel vector quantization network, incorporating meta-learning methods and VQNs.
2.	The approach is adaptable for multiple VQN models without affecting downstream tasks, so it could be straightforward to integrate into new pipelines.

Additional Weaknesses (in more detail):
1.	Computational Overhead: The new approach requires unrolled gradient steps (or finite difference approximations). The paper mentions an approximation but does not provide an in-depth breakdown of the training cost or memory usage on large codebooks.
2.	Limited Sensitivity Analysis: The ablation studies are somewhat narrow. The authors partially test turning off bi-level optimization or using various hyper-net types, yet do not systematically explore the effect of different unroll lengths or layer widths of the hyper-net.
3.	Generalization Beyond Vision: The paper focuses on visual tasks. It remains unclear for the effectiveness of other task losses and how they affect the codebook optimization by the hyper-net.

**Questions For Authors:**

1.	Choice of Unroll Steps: Have you tried multiple unroll steps (beyond one-step) or different finite-difference approximations? How does that affect training time and results?
2.	Potential Memory Overheads: Could you provide more precise measurements of how memory/time usage increases for the partial unrolling or the hyper-grad approximations?
3.	Domain Generalization: Do you think the approach can directly transfer to other discrete representation tasks such as speech tokens or molecular modeling, without major changes?

**Relation To Broader Scientific Literature:**

The authors reference VQVAE, VQGAN, and relevant codebook-improvement methods (Gumbel-VQ, VQGAN-LC, FSQ, etc.). They also draw comparisons to meta-learning methods and hyperparameter optimization.

**Theoretical Claims:**

The paper uses the logic of meta-learning, referencing the established notion that bi-level optimization can optimize both model and “hyperparameters” jointly.

---

> ### Author Rebuttal · Authors · 2025-04-01
>
> We appreciate your constructive feedback very much. We provide our response to your review as follows.
>
> > Computational Cost and Memory Overheads
>
>
> We conducted additional experiments to address your concerns. When evaluated on the CelebA dataset with a batch size of 128, the increase in memory usage is marginal and acceptable in practice. For time comparison, we set VQVAE as the baseline, which needs around 3.6 hours to finish the training of 50k steps (and does not improve after that). We find that, MQVAE only requires 2.4 hours to reach the same LPIPS score as VQVAE (around 9.5k steps), and can keep improving after that. This demonstrates that our method converges much faster than VQVAE and is able to outperform baselines with extended training time.
> | Method           | Memory (GB) | Wall time to reach baseline (h) | Total wall time (h) |
> | ---------------- | ----------- | ------------------------------- | ------------------- |
> | VQVAE (baseline) | 7.29        | 3.6                             | 3.6                 |
> | MQVAE            | 7.35        | 2.4                             | 12.2                |
>
>
> > Difficulty of converging using the meta-learning strategy
>
> Empirically, we did not observe stability issues during training. Theoretically, related convergence analyses for this type of gradient-based bilevel optimization algorithm can be found in [5], [6], and the references therein. Our MQ belongs to this type of optimization and is guaranteed to be stable and converge under certain conditions.
>
> > Demonstrate how codes are generated and selected and which part is optimized; The codebook distribution; Implementation Details
>
> Please follow this anonymous link https://anonymous.4open.science/r/MQVAE-B52C for the figure illustration and code implementation, and we will open-source them once the paper is accepted.
>
> > Compare with VQVAE+Affine+OPT+replace+l_2, and CVQ-VAE
>
> According to the performance reported in [1], MQVAE performs slightly worse than the combination of VQVAE+Affine+OPT+replace+$l_2$. Fortunately, we can show that when combined with additional techniques such as $l_2$ projection, MQVAE can still outperform [1] on the MNIST datasets, as shown in the table below. We also include a comparison with CVQ-VAE [2] for your reference.
>
> | Method                         | MSE ($\times 10^{-3}$) | LPIPS ($\times 10^{-1}$) |
> | ------------------------------ | ---------------------- | ------------------------ |
> | CVQ-VAE                        | 2.87                   | 3.73                     |
> | VQVAE+Affine+OPT+replace       | 1.81                   | 2.56                     |
> | VQVAE+Affine+OPT+replace+$l_2$ | 1.74                   | 2.27                     |
> | Ours+$l_2$                     | 1.64                   | 2.18                     |
>
> > Limited Sensitivity Analysis: Have you tried multiple unroll steps (beyond one step) or different finite-difference approximations? How does that affect training time and results?
>
> Our finite-difference (FD) approximation currently supports only one-step unrolling. We conducted additional experiments with alternative approximations, including the conjugate gradient (CG, [3]) and Neumann series (NMN, [4]) methods. Our ablation studies were done on the CelebA dataset. We found the type of approximation largely does not affect.
>
> | Method | MSE  | LPIPS |
> | ------ | ---- | ----- |
> | FD     | 3.10 | 0.14  |
> | CG     | 3.24 | 0.14  |
> | NMN    | 2.98 | 0.14  |
>
> > Downstream Utility and Domain Generalization: Do you think the approach can directly transfer to other discrete representation tasks, such as speech tokens or molecular modeling, without major changes?
>
> Based on our experiments, we believe that our framework can be directly adapted to other discrete representation tasks. Our approach is flexible in accommodating different types of task loss, as demonstrated in our experiments where both MSE loss and perceptual loss have been evaluated. In both cases, MQ shows superiority over VQ.
>
> [1] Huh, Minyoung, et al. "Straightening out the straight-through estimator: Overcoming optimization challenges in vector quantized networks." International Conference on Machine Learning. PMLR, 2023.
>
> [2] Zheng, Chuanxia, and Andrea Vedaldi. "Online clustered codebook." *Proceedings of the IEEE/CVF International Conference on Computer Vision*. 2023.
>
> [3] Rajeswaran, Aravind, et al. "Meta-learning with implicit gradients." *Advances in neural information processing systems* 32 (2019).
>
> [4] Lorraine, Jonathan, Paul Vicol, and David Duvenaud. "Optimizing millions of hyperparameters by implicit differentiation." *International conference on artificial intelligence and statistics*. PMLR, 2020.
>
> [5] Pedregosa, Fabian. "Hyperparameter optimization with approximate gradient." *International conference on machine learning*. PMLR, 2016.
>
> [6] Rajeswaran, Aravind, et al. "Meta-learning with implicit gradients." *Advances in neural information processing systems* 32 (2019).

---

> > ### Comment · Reviewer_VZz4 · 2025-04-05
> >
> > Thanks for the explanation.

---

### Official Review · Reviewer_yGdn · 2025-03-14

**Overall Recommendation:** 3

**Summary:**

This paper proposes a novel vector quantization training framework Meta-Quantization inspired by meta-learning, which decouples the optimization of codebook and autoencoder into two stages, enabling dynamic codebook generation and task-specific training. The proposed method outperforms existing vector quantization approaches on image construction and generation tasks.

**Claims And Evidence:**

This paper achieves direct backpropagation on the codebook instead of using STE, enabling codebook training task-specific.
The experiment in both image generation and reconstruction tasks shows that Meta-Quantization consistently outperforms multiple baselines and ablation methods, validating its superiority.
The experiment also shows that in codebook utilization,  the methods achieve the best performance.

**Essential References Not Discussed:**

Essential references have beed discussed.

**Experimental Designs Or Analyses:**

The experiments in this paper are comprehensive, conducting evaluations on state-of-the-art models VQVAE and VQGAN. The proposed method is compared against other SOTA models, incorporating codebook utilization comparisons alongside original evaluation metrics. Additionally, implicit codebook methods such as FSQ and LFQ are also analyzed. Experimental results demonstrate that the proposed approach not only efficiently utilizes the codebook but also achieves superior performance across all tasks.

In ablation studies, experiments are conducted to validate the effectiveness of bi-level optimization and compare different Hyper-Net architectures. Results showed that the MLP-based Hyper-Net outperformed other variants, confirming that more complex Hyper-Net designs consistently yield better performance.

**Methods And Evaluation Criteria:**

**Methods**

- The paper innovatively introduces a hypernet to take the place of the embedding-parameterized codebook. This substitution circumvents the need for direct optimization of the codebook itself. Moreover, after the first-stage training, only the generated codebook needs to be stored and can be directly applied in the subsequent training process.

- To tackle the optimization problems of the hypernet and the encoder-decoder, the paper employs a two-stage optimization framework. In this framework, the parameters of the two structures are optimized in a hierarchical manner, and an efficient gradient-based optimization algorithm with finite difference approximation is utilized.

  **However**, is the bi-level optimization approach adopted in this paper necessary? Since the Hyper-Network is also composed of linear layers or MLPs, if we perform backpropagation on the Hyper-network parameters $\psi$ together with the encoder-decoder parameters $\phi$ and $\theta$ using a single-step training method, would the performance differ significantly from bi-level optimization? Is there any practical or theoretical reason why bi-level optimization might yield superior results?

**Evaluation Criteria**

Evaluation with VQVAE

- In image reconstruction task, in addition to MSE and LPIPS, this paper uses the perplexity of the model as an evaluation metric to measure the similarity of the codebook. A higher perplexity value indicates a more uniform assignment of codes. The evaluation is carried out on the CIFAR10 and CelebA datasets.
- In image generation task, the FID evaluation metric is adopted. MaskGIT is applied to the CelebA dataset, and the results are extended to the image generation task, enabling the direct utilization of the codebook trained in the first stage.

Evaluation with VQGAN

- The model is trained on the ImageNet-1K and FFHQ datasets.
- In image reconstruction task, the evaluation metrics included rFID, LPIPS, PSNR, and SSIM. The assessment is conducted on the validation sets of ImageNet and FFHQ.
- In image generation task, the FID evaluation metric is used, and the evaluation is performed on the FFHQ dataset.

**Other Comments Or Suggestions:**

There are no other comments or suggestions.

**Other Strengths And Weaknesses:**

The description of convergence is inconsistent. While Figure 2 states that $\phi$ and $\theta$ are trained to convergence before training $\psi$, Algorithm 1 shows that they are updated together. This discrepancy creates ambiguity regarding the actual optimization procedure implemented in the paper.

**Questions For Authors:**

In the introduction, it is mentioned that the codebook utilization in previous methods is low. However, in the experiments (Table 3, 4, 5), the codebook utilization of VQGAN-LC is also quite high. Please provide a justification for this conclusion and explain the advantages of this method over VQGAN-LC in terms of codebook utilization.

**Relation To Broader Scientific Literature:**

This paper adopts the DART method in the field of vector quantization, which has implications for the image tokenization domain but has no impact on other areas.

**Theoretical Claims:**

This paper references the theoretical analysis of gradient analysis for DART's two-level optimization and generalizes the optimization algorithm, containing no errors in proofs.

---

> ### Author Rebuttal · Authors · 2025-04-01
>
> We appreciate your constructive feedback very much. We provide our response to your review as follows.
>
> > Is the bi-level optimization approach adopted in this paper necessary?
>
> Yes, it is necessary. The two components of our method address distinct challenges. Specifically, the hypernet resolves the issue of codebook collapse and enhances codebook utilization, while the bilevel optimization approach ensures that the task loss gradient reaches the codebook. As demonstrated in our ablation studies (Section 5.3), both components are essential and must be combined to achieve optimal results.
>
> > The description of convergence is inconsistent.
>
> We apologize for the confusion. The figure illustrates the workflow of bilevel optimization for a single gradient step, whereas the algorithm blocks detail our specific implementation. Our approach applies a finite difference-based approximation, wherein one gradient descent step approximates the converged solution of the autoencoder.
>
> Thus, a consistent pseudo-algorithm is to replace
>
> "Update $\psi$ using gradient descent:  $\nabla_{\psi}\mathcal{L}(\phi-\xi\nabla_\phi\mathcal{L}(\phi, \theta, \psi), \theta-\xi\nabla_\theta\mathcal{L}(\phi, \theta, \psi), \psi)$" in the while loop with
>
> "Copy $\phi^\prime=\phi$ and $\theta^\prime=\theta$, and update $\phi^\prime$ and $\theta^\prime$ using gradient descent by $\nabla_\phi\mathcal{L}(\phi, \theta, \psi)$ and $\nabla_\theta\mathcal{L}(\phi, \theta, \psi)$ until converging, resulting in $\phi^{\prime*}$ and $\theta^{\prime*}$ (the computation graph is retained during descent). Update $\psi$ using gradient descent:  $\nabla_{\psi}\mathcal{L}(\phi^{\prime*}, \theta^{\prime*}, \psi)$". This is consistent with the figure description, in which the lower level is optimized until convergence, followed by one update of the upper level. We use a one-step unroll scheme, approximating $\phi^{\prime*} \approx \phi-\xi\nabla_\phi\mathcal{L}(\phi, \theta, \psi)$ and $\theta^{\prime*} \approx \theta-\xi\nabla_\theta\mathcal{L}(\phi, \theta, \psi)$, which corresponds to the pseudo-algorithm presented in our paper.
>
> > Justification of advantages over VQGAN-LC.
>
> VQGAN-LC also addresses codebook under-utilization and thus exhibits higher codebook usage compared to the original VQGAN. In contrast, our approach employs a hypernet reparameterization, which enables simultaneous updates of all codebook entries.

---

### Official Review · Reviewer_bZLB · 2025-03-15

**Overall Recommendation:** 3

**Summary:**

The paper proposes to train VQ-VAE under a meta-learning framework. To be more specific, the paper introduces a hyper-network to replace the embedding-parameterized and trains the model with bi-level optimization. Experiments are conducted on image reconstruction and generation tasks. The proposed MQ-VAE improves over the VQ baseline.

**Claims And Evidence:**

The Gradient Analysis section is quite interesting. However, empirical evidence is lacking. I can only find an ablation in Table 6, which disables the indirect gradient by zeroing out $\xi$.  However, the improvement is not much. I am curious about the true magnitude of the indirect gradient compared to the direct one. A plot wrt the training iterations may better support this.

**Essential References Not Discussed:**

See above.

**Experimental Designs Or Analyses:**

The experimental designs are sound and comprehensive. Specifically, the baseline methods include VQVAE and VQ-GAN.

**Methods And Evaluation Criteria:**

++ Applying the bi-level optimization to training VQ-VAE/VQGAN is reasonable.

++ The evaluation criteria include image reconstruction and image generation.

**Other Comments Or Suggestions:**

No.

**Other Strengths And Weaknesses:**

==== Strength ====

1. The paper is well-written and easy to understand. The overview figure nicely illustrates the gradient flow.

2. Evaluations are comprehensive.

==== Weaknesses ====

Some weaknesses have been covered above. There is one more weakness that I would love to point out.

1. Bi-level optimization introduces the computational cost overhead. The paper didn't show any result on the computational cost. There are not many implementation details about the paper. Nevertheless, an interesting experiment is to see the comparison between MQ-VAE and regular single-level optimized VQVAE with the same training budget. In other words, the vanilla VQVAE would enjoy a longer training schedule, which will generally lead to constant improvement from my experience.

**Questions For Authors:**

1. Training cost compared to vanilla VQ-VAE

2. A more fair comparison with VQ-VAE under the same training budget. In other words, the authors should allow the baseline to train for longer iterations.

3. More illustrations on the gradient analysis (see the claims and evidence section).

**Relation To Broader Scientific Literature:**

The difficulty of training VQ-VAE lies in the non-differentiability of the *argmin* operator in the bottleneck. This paper proposes a bi-level optimization approach to mitigate this issue. A recent paper [1] tackles this by bounding the quantization error with a spherical vector quantization-like bottleneck. The vector quantization is implicitly modeled by a linear projection, which resembles the hypernetwork design. However, the model can be trained without bi-level optimization, achieving very similar results on ImageNet 128x128 in Table 4. It looks like these two papers try to tackle the same problem from different perspectives, and it would be good to see if [1] can also benefit from the proposed bi-level optimization technique. [2] also proposes to propagate gradients more smoothly via a rotation.

[1] Zhao, et al. "Image and video tokenization with binary spherical quantization." arXiv preprint arXiv:2406.07548 (2024).

[2] Fifty, et al. "Restructuring Vector Quantization with the Rotation Trick." arXiv preprint arXiv:2410.06424 (2024).

**Theoretical Claims:**

The formulation and derivations look correct. There are no particular proofs for theoretical claims.

---

> ### Author Rebuttal · Authors · 2025-04-01
>
> We appreciate your constructive feedback very much. We provide our response to your review as follows.
>
> > Magnitude Comparison
>
> In our experiment, we have found that the magnitude of both indirect and direct are around $10^{-2}$, so none of them dominate each other. Please follow this anonymous link https://anonymous.4open.science/r/MQVAE-B52C for the figure.
>
> > Whether [1] can benefit from bi-level optimization
>
> We want to clarify that [1] addresses an essentially different problem. In our formulation, the core idea is to generate a codebook using (1) a learnable embedding and (2) a learnable transformation (e.g., linear projection). The method presented in [1] does not involve learnable embeddings; moreover, its projections are considered part of the backbone autoencoder architecture, whereas in our case, the projection (a hypernet) is integrated into the codebook. Thus, [1] falls outside the scope of our study, and we cannot comment on applying bilevel optimization to [1] based on our main paper's arguments.
>
> >  Relation to [2]
>
> This work primarily focuses on improving the STE estimator used in the original VQVAE paper. Although it facilitates smoother gradient propagation through the non-differentiable quantization layer, the codebook update still depends solely on the distribution of codes and features and does not incorporate the task loss. Therefore, that work addresses a different issue and is orthogonal to our approach.
>
> > Computation Cost and Implementation details
>
> We conducted additional experiments to address your concerns. When evaluated on the CelebA dataset with a batch size of 128, the increase in memory usage is marginal and acceptable in practice. For time comparison, we set VQVAE as the baseline, which needs around 3.6 hours to finish the training of 50k steps (and does not improve after that). We find that, MQVAE only requires 2.4 hours to reach the same LPIPS score as VQVAE (around 9.5k steps), and can keep improving after that. This demonstrates that our method converges much faster than VQVAE and is able to outperform baselines with extended training time.
> | Method           | Memory (GB) | Wall time to reach baseline (h) | Total wall time (h) |
> | ---------------- | ----------- | ------------------------------- | ------------------- |
> | VQVAE (baseline) | 7.29        | 3.6                             | 3.6                 |
> | MQVAE            | 7.35        | 2.4                             | 12.2                |
>
> [1] Zhao, et al. "Image and video tokenization with binary spherical quantization." arXiv preprint arXiv:2406.07548 (2024).
>
> [2] Fifty, et al. "Restructuring Vector Quantization with the Rotation Trick." arXiv preprint arXiv:2410.06424 (2024).

---

### Official Review · Reviewer_e9kM · 2025-03-18

**Overall Recommendation:** 3

**Summary:**

This paper introduces Meta-Quantization (MQ), by using a hyper-net and bi-level optimization to alternatively train the codebook with the autoencoder in Vector Quantization Networks (VQN). Experiments show MQ has better codebook ultilization, image reconstrcution and generation performance.

**Claims And Evidence:**

Yes. VQ has optimization issues and this paper tackles this.

**Essential References Not Discussed:**

No.

**Experimental Designs Or Analyses:**

Lack of comparison on the training stability and efficiency of the optimization with other methods.

**Methods And Evaluation Criteria:**

- The optimization seems to be very complicated. Why not use simpler method such as clustering the embeddings or EMA update? What are the advantages against existing methods for better codebook ultilization?
- What is the training stability and efficiency of the optimization compared to other methods?
- I am confused about the details of the hyper-net. In lines 287-292, it is described as an MLP. Does it mean it takes a single embedding and generate a single code entry? How does it produce the whole codebook?

**Other Comments Or Suggestions:**

Please provide more details of the hyper-net.

**Other Strengths And Weaknesses:**

Strengths:
- The MQ method and the optimization algorithm makes sense.
- Results are strong.

Weekness:
- It is not clear if the training is efficient.

**Questions For Authors:**

In lines 69-74, what is the "specific task"? What is "task-aware"? Is it the VQVAE loss?

**Relation To Broader Scientific Literature:**

This paper is related. It proposes a meta-learning approach for codebook optimization.

**Theoretical Claims:**

Yes.

---

> ### Author Rebuttal · Authors · 2025-04-01
>
> We appreciate your constructive feedback very much. We provide our response to your review as follows.
>
> > Why not use a simpler method such as clustering the embeddings or EMA update
>
> One of the advantages and novel aspects of MQ, compared to simpler methods, is that the codebook update follows a more complete loss path in that it directly interacts with the task loss gradient.
>
> For example, the simplest VQVAE is designed to reconstruct the input image using the mean squared error loss. However, updating the codebook by enclosing the code and feature in latent space does not involve the MSE between the image and the reconstructed output. In contrast, our meta-learning-based method avoids this incomplete gradient issue. Please refer to Figure 2 in the main paper for additional details.
>
> > What is the training stability and efficiency of the optimization compared to other methods?
> >
>
> Empirically, we did not observe stability issues during training. Theoretically, related convergence analyses for this type of gradient-based bilevel optimization algorithm can be found in [1], [2], and the references therein. Our MQ belongs to this type of optimization and is guaranteed to be stable and converge under certain conditions.
>
> > It is not clear if the training is efficient.
>
> We conducted additional experiments to address your concerns. When evaluated on the CelebA dataset with a batch size of 128, the increase in memory usage is marginal and acceptable in practice. For time comparison, we set VQVAE as the baseline, which needs around 3.6 hours to finish the training of 50k steps (and does not improve after that). We find that, MQVAE only requires 2.4 hours to reach the same LPIPS score as VQVAE (around 9.5k steps), and can keep improving after that. This demonstrates that our method converges much faster than VQVAE and is able to outperform baselines with extended training time.
> | Method           | Memory (GB) | Wall time to reach baseline (h) | Total wall time (h) |
> | ---------------- | ----------- | ------------------------------- | ------------------- |
> | VQVAE (baseline) | 7.29        | 3.6                             | 3.6                 |
> | MQVAE            | 7.35        | 2.4                             | 12.2                |
>
> > I am confused about the details of the hyper-net. In lines 287-292, it is described as an MLP. Does it mean it takes a single embedding and generates a single code entry? How does it produce the whole codebook?
>
> In this case, the MLP will separately project each codebook entry with respect to their dimensionality, meaning if the embedding is $e_n, n=1...,N$, the generated codebook is $MLP(e_n), n=1,...,N$. Nevertheless, if any $MLP(e_n)$ is selected and receives a gradient, the MLP will be updated, resulting in a different codebook being generated by this updated MLP.
>
> > In lines 69-74, what is the "specific task"? What is "task-aware"? Is it the VQVAE loss?
>
> The specific task depends on the benchmark. For instance, in image reconstruction tasks, the task loss is the MSE loss in the case of VQVAE or a combination of perceptual loss, MSE loss, and adversarial loss in the case of VQGAN.
> Task awareness means that the gradient used to update the codebook directly incorporates the task loss. One contribution of this work is to establish this connection. In Figure 2, the task loss reaches the codebook through various pathways; previous VQ methods updated the codebook solely based on the gradient of selected codes toward selected features, without incorporating the task loss.
>
> [1] Pedregosa, Fabian. "Hyperparameter optimization with approximate gradient." *International conference on machine learning*. PMLR, 2016.
>
> [2] Rajeswaran, Aravind, et al. "Meta-learning with implicit gradients." *Advances in neural information processing systems* 32 (2019).

---

### Decision · Program_Chairs · 2025-05-01

**Decision:**

Accept (poster)

**Comment:**

All of reviewers lean toward acceptance post-rebuttal. The AC checked all the materials and concurs that the paper has proposed a reasonable and principled approach to tackle the difficulty of training online vector-quantized tokenizers for vision. While there are limitations (e.g., the overall complexity of the bi-level framework), the work presents a valuable attempt to mitigate the issue of training vector-quantized networks, and therefore should be accepted. Please incorporate necessary changes in the final version, and open-source code/models as promised.